# Graphene Oxide-Doped Polymer Inclusion Membrane for Remediation of Pharmaceutical Contaminant of Emerging Concerns: Ibuprofen

**DOI:** 10.3390/membranes12010024

**Published:** 2021-12-25

**Authors:** Abdul Latif Ahmad, Oluwasola Idowu Ebenezer, Noor Fazliani Shoparwe, Suzylawati Ismail

**Affiliations:** 1School of Chemical Engineering, Engineering Campus, Universiti Sains Malaysia, Nibong Tebal 14300, Pulau Pinang, Malaysia; oluwasolaebenezer@student.usm.my (O.I.E.); chsuzy@usm.my (S.I.); 2Food Technology Department, The Federal Polytechnic Ado Ekiti, Ado Ekiti 360231, Ekiti State, Nigeria; 3Faculty of Bioengineering and Technology, Universiti Malaysia Kelantan (UMK), Jeli Campus, Jeli 17600, Kelantan, Malaysia; fazliani.s@umk.edu.my

**Keywords:** Ibuprofen, contaminants of emerging concerns, polymer inclusion membrane, graphene oxide, chemical stability, wastewater, remediation

## Abstract

The application of polymer inclusion membranes (PIMs) for the aquatic remediation of several heavy metals, dyes, and nutrients has been extensively studied. However, its application in treating organic compounds such as Ibuprofen, an emerging pharmaceutical contaminant that poses potential environmental problems, has not been explored satisfactorily. Therefore, graphene oxide (GO) doped PIMs were fabricated, characterized, and applied to extract aqueous Ibuprofen at varied pH conditions. The doped PIMs were synthesized using a low concentration of Aliquat 336 as carrier and 0, 0.15, 0.45, and 0.75% GO as nanoparticles in polyvinyl chloride (PVC) base polymer without adding any plasticizer. The synthesized PIM was characterized by SEM, FTIR, physical, and chemical stability. The GO doped PIM was well plasticized and had an optimal Ibuprofen extraction efficiency of about 84% at pH of 10 and 0.75% GO concentration. Furthermore, the GO doped PIM’s chemical stability indicates better stability in acidic solution than in the alkaline solution. This study demonstrates that the graphene oxide-doped PIM significantly enhanced the extraction of Ibuprofen at a low concentration. However, further research is required to improve its stability and efficiency for the remediation of the ubiquitous Ibuprofen in the aquatic environment.

## 1. Introduction

A composite material consists of two or more components designated as matrix and filler. The primary function of the filler is to fill in the matrix and synergistically combine the advanced properties of the matrix and filler for improved performance, as evidenced by the several significant breakthroughs and considerable benefits for commercial water treatment and desalination applications [1]. Over the past few decades, nanoparticles have been incorporated into several industrial processes and applications, particularly carbon-based materials used as charges in water treatment mechanisms [2,3]. For example, several authors [2,3,4,5,6,7,8,9,10] have reported graphene and its derivatives as suitable fillers to remove several organic and inorganic wastewater contaminants with remarkable efficiencies [11,12].

Following its discovery by Novoselov et al. in 2004, graphene, a single-layer 2-D structure material with thin thickness at the atomic level, has received serious attention in the broad scientific community and is notably recognized as a major potential component in next-generation membrane technologies (Graphene-Info, 2012), [13,14,15,16,17,18]. Graphene has good water channel properties, large surface area, remarkable tensile strength (~1 Tpa), the exceptional electrical conductivity of ~106 s cm^−1^ at ordinary temperature, high thermal conductivity of ~5000 W m K^−1^, enhanced active sites, large delocalized π-electron systems, and chemical stability coupled with an exclusive tunable size structure. Hence, the material can be easily engineered for different aqueous filtration techniques. Furthermore, graphene is an ideal adsorbent for wastewater making it a unique nanofiller material [12,13,18].

Native graphene is hydrophobic, which limits its application in wastewater treatment processes. However, the oxidized form of graphene (GO), formed from thermal or chemical exfoliation of graphite oxide, exhibits hydrophilic properties and is frequently used as nanofillers for water remediation mechanisms. Furthermore, the presence of oxygenated functional groups such as the carboxyl, the primary hydroxyl group on edge plane, epoxy, tertiary hydroxyl, and carbonyl groups on basal planes, among others found in GO, coupled with their excellent solubility and the potential for surface functionalization, make GO a better nanocomposite material compared to the native graphene. In addition, previous reports have proven the remarkable removal response of magnetic/graphene nanoparticles or GO composites for contaminants such as chromium, copper, arsenic, cadmium, lead, cobalt, and an organic dye [19].

The advent of new and more sensitive analysis methods has made it possible to detect very low concentrations of various potentially dangerous contaminants in the aquatic systems up to ng/L detection level. This newly detected and potentially harmful chemical class is now globally classified as “emerging pollutants” or Contaminants of Emerging Concern (CECs). According to Poynton and Robinson, the US Geological Survey (2016) has defined CECs as “any unregulated synthetic or naturally occurring chemical or any microorganism not commonly assessed in the environment but with the potential of been dispersed in the environment causing known or suspected adverse ecological and (or) human health effects [19,20,21,22,23].

Pharmaceutical and related compounds, otherwise known as active pharmaceutical ingredients (APIs), are now considered emerging environmental contaminants. Such a wide range of emerging pharmaceutical contaminants found in wastewater include analgesics and anti-inflammatory drugs (Naproxen, Ketoprofen, Ibuprofen, Diclofenac, Indomethacin, Acetaminophen, Mefenamic acid, Propyphenazone), Anti-ulcer agents (Ranitidine), Psychiatric drugs (Paroxetine), Antiepileptic drugs (Carbamazepine), Antibiotics (Ofloxacin, Sulfamethoxazole, Erythromycin), B-blockers (Atenolol, Metoprolol), Diuretics (Hydrochlorothiazide), Hypoglycaemic agents (Glibenclamide), lipid regulator, and cholesterol-lowering statin drugs (Gemfibrozil, Bezafibrate, Clofibric acid, Pravastatin) [21,24,25,26,27,28,29]. For example, Ibuprofen, a typical member of the nonsteroidal anti-inflammatory drugs (NSAIDs) listed in Essential Drugs lists by the World Health Organization (WHO) in 2010, is one of the most widely used drugs globally and the third most popular, highly prescribed, and most salable over-the-counter medicine in the world. Ibuprofen is used for the medicament of myoskeletal injuries, rheumatoid arthritis, and fever; hence, it is one of the frequently detected pharmaceutical CECs [24,30,31,32,33,34]. Salgado et al. [32] have reported up to 11.5 µg/L in the influent, 0.9 µg/L in the effluent of some musks, and up to 22.6 µg/g in the sludge of five wastewater treatment plants (WWTPs) in Portugal.

Therefore, the concentration and toxicity of Ibuprofen in wastewater treatment plants and water bodies are a growing concern. According to Marchlewicz et al., the increasing Ibuprofen pollutant in the aquatic environment poses a potential hazardous impact due to its bioactive nature and ubiquitous presence in surface water, rivers and lakes, hospital, and municipal water, and probably in drinking water. Unfortunately, conventional water treatment cannot eliminate Ibuprofen efficiently [29,30,31].

Polymer inclusion membranes (PIMs) have been widely studied and reported to be efficient in removing dyes, heavy metals, and nutrients from aquatic systems. However, only a few studies have investigated the removal of aqueous Ibuprofen using PIM. Hence, this study evaluates the stability and potential of functionalized polymer inclusion membrane to remove low concentration Ibuprofen, which prolonged accumulation has been recognized as a pharmaceutical’s endocrine disruptors in the aquatic system [35,36,37,38].

## 2. Materials and Methods

### 2.1. Materials

Ibuprofen, IBP (purity ≥ 99.0%) (Sigma–Aldrich, St. Louis, MO, USA) without further purification, was used to simulate IBP aqueous pollutant model. Poly(vinyl chloride) (Mw~43000) as Polymers, Tetrahydrofuran (THF) (99.9%) (Darmstadt, Germany) as solvent was purchased from Sigma–Aldrich, Graphene (OH) (multi-layered) (Purity: 99.99%) of lateral Size: 10 μm (India) as nanoparticles and Aliquat^®^ 336 TG obtained from Alfa Aesar Thermofisher Scientific (Heysham, Lancashire, UK) as ionic exchange carrier. The reagents used include 0.1M Hydrochloric acid (Merck, Darmstadt, Germany) and 0.1 M Sodium hydroxide. All reagents employed are of standard analytical quality.

### 2.2. Preparation of the Pristine and Control Polymer Inclusion Membrane

Polymer (PVC) was dissolved in THF solvent and stirred to form a homogenous dope solution at 65 °C. The pristine (PIM GOO) was fabricated without adding carrier or graphene oxide. The same step was followed to fabricate the control PIM (PIM GO) but with an Aliquat carrier (Formulation according to Table 1). The homogenous solution was then sonicated to remove any air bubbles. Subsequently, the dope solution formed was then poured into a smooth glass plate pre-set at 30 µm thickness in an automatic film machine filmograph (K4340 Automatic Film Applicator, Elcometer) US. The cast membrane was then left at room temperature to allow the THF to evaporate before peeling off, and subsequent drying time beyond 24 h was allowed.

### 2.3. Preparation of the Graphene Doped Polymer Inclusion Membrane

Polymer (PVC), Aliquat 336 carrier, and an appropriate varied amount of graphene oxide were dissolved in THF (according to Table 1) under continuous magnetic stirring at 65 °C until a homogeneous solution was obtained. The solution was then sonicated to remove any air bubbles. Subsequently, the dope solution formed was then poured into a glass plate pre-set at 30 µm thickness in an automatic film machine filmograph (K4340 Automatic Film Applicator, Elcometer) (Manchester, UK). The cast membrane was then left at room temperature to allow the organic solvent to evaporate for 24 h, after which the formed thin-sheet graphene doped PIM was peeled off and allowed for further drying for at least another 24 h.

### 2.4. Membrane Characterization

#### 2.4.1. Scanning Microscopy Analysis

The surface morphology of the PIMs was observed by scanning electron microscopy (SEM) HITACHI (TM3000) Tokyo, Japan, at 5 kv and ×100 magnification. The membrane samples were initially gold-plated using Quorum SC7620 (Laughton, East Sussex, USA). 

#### 2.4.2. Fourier Transform–Infrared Spectroscopy

Fourier transform–infrared spectroscopy (FT–IR) measurements were performed on the fabricated and tested PIMs. The test was conducted to identify the interrelations among the PVC polymer, aliquat 336 carrier, and the GO nanofiller [39]. Therefore, the IR spectra of the fabricated PIMs were determined by Fourier transformed infrared spectrometry (Thermo Scientific, Nicolet is10, Waltham, MA, USA) to identify the functional group in the Doped-PIMs. Data were collected in transmission mode after 32 scans of each membrane in wave numbers from 4000 to 500 cm^−1^.

#### 2.4.3. Membrane Stability

##### Physical and Chemical Stability Assessment

Physical stability is one of the most significant characteristics of a PIM. A physically stable PIMs should be able to bend without tearing or visible deformation, signaling an excellent physical strength [40]. Therefore, the fabricated PIMs were subjected to physical tests by manually bending and stretching the membrane sheets at a minimal force.

The chemical stability of the polymer inclusion membrane, particularly in environmental conditions of interest, is a critical factor for PIM’s practical application in water remediation. The loss of carrier during application attributed to the weight loss of the PIM after being subjected to specific environmental conditions is a helpful tool to assess the membrane’s chemical stability [40,41,42,43]. Therefore, the chemical stability of the optimum fabricated graphene doped PIMs was investigated and compared to the control PIM. The optimum GO doped-PIM and control PIM membranes with an average weight of 0.0171 ± 0.002 g was cut out for the investigation by monitoring the mass loss after 6 h in contact with 0.1 M NaOH solution (pH~10), 0.1 M HCL solution (pH~1), and phosphate buffer solution (pH~6.8) and continued stirring at ~400 rpm. The membranes were subsequently dried at room temperature until a constant weight was obtained. The percentage mass loss was calculated using Equation (1): All measurements were conducted in triplicate.
(1)$$\mathrm{Mass}\;\mathrm{loss}\%=\frac{W_{o}-W_{t}}{W_{o}}\times 100\;$$
where *W_o_* is the initial weight of the PIM, and *W_t_* is the final PIM’s constant weight after drying.

### 2.5. Performance Extraction of Pharmaceuticals Ibuprofen

This test was performed to investigate the removal efficiency of the fabricated graphene oxide doped PIMs for a 10 mg/L simulated aqueous Ibuprofen solution. Each PIM understudy had an approximate diffusing area of 7.07 cm^2^ and clamped between two 250 mL identical diffusion cells on a Teflon ring-shaped support (Figure 1). The feed compartment was filled with 10 mg/L aqueous Ibuprofen prepared serially from a stock solution of 20 mg/L by diluting it with deionized water. The feed solution phase was subsequently adjusted to a pH range of 2, 6, and 10 using 0.1 M HCL and 0.1 M NaOH as appropriate. The receiving phase compartment was filled with 0.1 M HCL. Each phase was simultaneously stirred at a speed of approximately ~400 rpm with the aid of a magnetic bar at room temperature. The Ibuprofen concentration was determined by U-Vis spectrometry (Spectroquant Pharo 300, Merck, Darmstadt, Germany) in a quartz cuvette at 222.5 [44,45], which corresponds to the maximum Ibuprofen absorbance detected in the present study and conditions [46,47]. The percentage Ibuprofen extraction (E%) was calculated using Equation (2)
(2)$$E\%=\frac{Con.\;feed,0-Con.\;feed,0}{Con.feed,0}\times 100$$
where *Con. Feed*,0 is the initial concentration of Ibuprofen in the solution, *Con. Feed*, *t* is the Ibuprofen concentration in the solution after equilibrium is reached.

## 3. Results

### 3.1. Characterization of Fabricated Graphene Doped Polymer Inclusion Membranes

#### 3.1.1. Scanning Electronic Microscopy

SEM images of the surface of all the fabricated PIMs are presented in Figure 2. A homogeneous microstructure of the PIM surface indicating a uniform distribution of graphene oxide and Aliquat carrier in the PVC polymer matrix was observed. Therefore, the fabricated PIMs can be classified as a dense thin film with no apparent porosity probably because of the low Aliquat 336 carrier used in its formulation because according to Xu et al., [48], a low Aliquat 336 content (i.e., below 30%) shows micropores features with the extractant molecules probably “entangled” within the polymer chains making the extractant molecules behaves like a plasticizer but with low activity and mobility through the polymer matrix. The dispersed GO within the surface of the PIMs G1–G3 tends to be uniformly grafted with the formation of apparent tiny pores that are less than a few µm and therefore could not be easily detectable by the SEM acquisitions.

#### 3.1.2. FT–IR Spectroscopy

FT–IR analysis is an essential tool to simultaneously determine the organic components, chemical bond, and organic content based on the nature of the chemical interactions between different components used in the fabrication of PIMs [39,43,49,50]. The FTIR results obtained for the fabricated PIMs (GOO-G3) and the optimum PIM G3 before and after extraction are presented in Figure 3 and Figure 4, respectively.

The pure PVC in the pristine (PIM GOO), control PIM (GO), and graphene oxide doped PIMs (G1-G3) is detected at the spectrum bands of 684, 685, and 686 cm^−1^, which is attributable to the bending modes of the C−H bonds, while the bands at 832, 1250, and 1330 cm^−1^ correspond to the stretching modes of the C−Cl bonds. The band at 1430 cm^−1^ and 2910 to 2970 cm^−1^ is attributed to the C−H bonds stretching. The C−C stretching bond of the PVC backbone chain occurs at 1060 and 1090 cm^−1^, while the C−H trans 4 wagging modes occur at 957 cm^−1^ [39,51,52].

The control PIM GO and the graphene doped PIMs G1–G3 distinctly revealed the Aliquat carrier added to them without any significant modification in the polymer chain of the PVC polymer. The quaternary amine group (CH2–N and CH3–N) of the Aliquat 336 carrier showed a peak at 2850 cm^−1^ [51,53]. A new absorption peaks band was also observed at the spectrum of 3380 and 3390 cm^−1^, indicating the CN group, which is unique to the Aliquat-336, confirming the physical integration of the Aliquat carrier within the PIMs’ matrix.

The presence of graphene oxide in PIMs’ G1–G3 can be observed by the ring structure of the GO as indicated by the peak at 2160 cm^−1^ [54,55]. Similarly, the peaks at 3380 cm^−1^ represent the O-H stretching vibration of hydroxyl groups, while the peak of 1630 cm^−1^ is the C=O unit of -COOH groups stretch band of graphene oxide [56,57].

To evaluate the mechanism of Ibuprofen removal by the optimum graphene doped PIM and at the optimum pH, a comparative infrared spectrum of PIM G3 before and after Ibuprofen extraction was investigated, as shown in Figure 4. A new distinct peak at 1710 cm^−1^, recognized as stretching C=C, indicates the Ibuprofen complex on the graphene oxide-doped PIM [57,58]. The peaks at 3380 cm^−1^ and 2160 cm^−1^ representing the ring structure of the GO are no longer seen, indicating a likely loss of the nanoparticles due to leaching occasioned by the instability of nanoparticle engrafted membranes [58].

#### 3.1.3. Physical and Chemical Stability

PIMs are usually considered physically stable when bent without tearing or visibly deforming the thin membrane structure. The results of the physical stability assessment to determine the effect of incorporated GO nanoparticles on the transparency and flexibility of the fabricated PIMs are presented in Table 2. It was observed that the incorporation of graphene oxide successfully plasticized the doped PIMs in the absence of a plasticizer, making them supple and flexible. In contrast, the pristine and the control PIM with 10% Aliquat carrier were brittle and unamenable to moderate stretching and bending. This result shows that the plasticized GO-doped PIM decreased the rigidity of the three-dimensional structure of the PVC polymer used, thereby enhancing the deformation, durability, workability without rupturing, and transport efficiency [59,60].

A critical and well-known drawback of PIM is its chemical instability due to leaching out of the carrier from the membrane into the aqueous solution. The rate of mass loss of the membrane under a specific condition, such as the pH of the media and membrane constituents, is strongly related to the loss of efficiency of the membrane during application [41,61]. The result obtained for the chemical stability of the optimum PIM (G3) and the control PIM (GO) in acidic, neutral, and alkaline solutions is presented in Figure 5. The results showed that the PIMs’ stability is generally affected by the pH of the medium. There is greater instability in the alkaline (AL) medium than the acidic (AC) and neutral (NT) media, which is similar to the finding of Kunène et al. [40] and Moulahcene et al. [43]. The mass loss of 10.256 ± 0.581% in the alkaline medium indicates the high instability of G3; this was also confirmed by the loss of some GO functional groups in the PIM after Ibuprofen extraction as revealed by the FTIR (Figure 4). The loss of nanoparticles due to material leaching out of the membrane matrix is a common limitation, affecting nanoparticles’ enhanced membrane [11]. This observation is probably due to the high hydrophilic nature of GO [62,63] and the PIM’s thickness (30 µm) in this work. Although thin PIM’s thickness enhanced extraction rate, it is also prone to leaching compared to a thicker membrane [40,64,65,66]. Hence, further modifications may be required to enhance the stability of the doped PIM but without compromising the precise separation performance of the membrane [67,68]. However, the mass loss of the control PIM (GO) in the alkaline medium (12.121 ± 0.548%) is higher than that of G3, indicating relatively improved stability of the latter. The mutual chemical interaction between the positive charge of the cationic Aliquat 336 carrier and concentrated hydroxide (OH−) anions from the NaOH and the dehydrochlorination reaction between the poly (chloroethene) in the PVC and quaternary ammonium compounds such as the Aliquat 336 have been attributed to the easier leaching out of aliquat carrier in an alkaline media compared to the acidic media [40,69,70].

### 3.2. Effect of Amount of Doped GO and Feed pH on the Efficiencies of Ibuprofen Removal

The effect of the different amounts of GO added to the doped PIMs (0, 0.15, 0.45, 0.75) compared with the pristine PIM was evaluated using 10 mg/L aqueous Ibuprofen extraction performance. Additionally, the effect of pH of the feed solution was also assessed using the optimal PIM.

#### 3.2.1. Effect of Graphene Oxide Concentration on Ibuprofen Extraction

The results of the efficiency of Ibuprofen extraction using the fabricated PIMs are as shown in Figure 6. Removal efficiency increased significantly between the doped PIMs and the control PIM GO after 96 hr. All the GO-doped PIMs showed good extraction performance. However, PIM G3 gave the highest removal efficiency of ~77%. The least value of ~70% removal efficiency obtained in the present work is greater than the optimum value of ~55% obtained by Moulahcene et al. [43] using PIM consisting of an insoluble β-CD polymer as a carrier in the presence of dibuthylphtalate (DBP) as a plasticizer. This study established that graphene oxide doped PIM with Aliquat 336 exchange carrier can effectively extract emerging Ibuprofen contaminant from aqueous water as opined by [71]. The present finding also confirmed the claim that the good negative charge surface area of GO and its electrostatic interactions are ideal sorbents, particularly for aromatic ring molecules such as the Ibuprofen drug [13,16,18,72,73,74].

#### 3.2.2. Effect of pH of Feed Solution on Ibuprofen Extraction

To investigate the effect of pH on the removal efficiency of Ibuprofen, the pH of the feeding solution was adjusted to 2, 6, and 10 (Figure 7). It is well known that the pH of feed solution significantly affects the removal effectiveness of pharmaceutical contaminants using membranes [30]. The result shows an extraction efficiency of ~84%, ~83%, and ~77% at the feed solution pH of 10, 6, and 2, respectively. The results indicate that the extraction efficiency increased when the pH of the feed solution increased. However, the effect of feed solution pH on Ibuprofen extraction efficiency using GO-dosed PIM is not very pronounced in the present study. The maximum extraction efficiency of ~84 percent occurs at a pH of 10. 

## 4. Conclusions

The effect of GO enhanced PIM by incorporating a small amount of GO as nanoparticle and Aliquat carrier on the remediation of trace Ibuprofen (a pharmaceutical contaminant of emerging concerns) and membrane stability was successfully studied. The fabricated GO-doped PIM was more stable in acidic and neutral media than in the alkaline medium. Furthermore, the optimum extraction efficiency of ~84% obtained indicates a high potential for Ibuprofen extraction using GO doped PIM. However, further research is required to improve the stability of the GO doped PIM and to study the effect of other operating conditions, such as the feed concentration and receiving phase types.

## Figures and Tables

**Figure 1 membranes-12-00024-f001:**
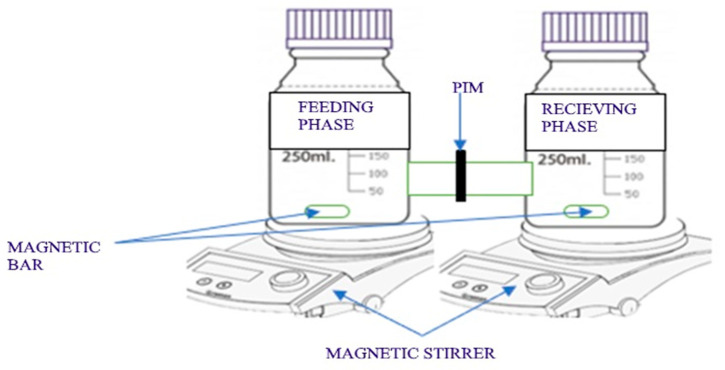
Set-up of the identical diffusion cell on a Teflon ring-shaped support PIM.

**Figure 2 membranes-12-00024-f002:**
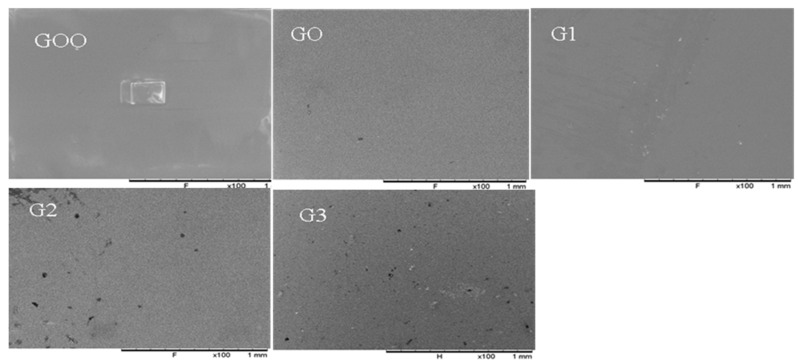
SEM Image of PIM GOO, GO, G1, G2, and G3 × 100 magnification.

**Figure 3 membranes-12-00024-f003:**
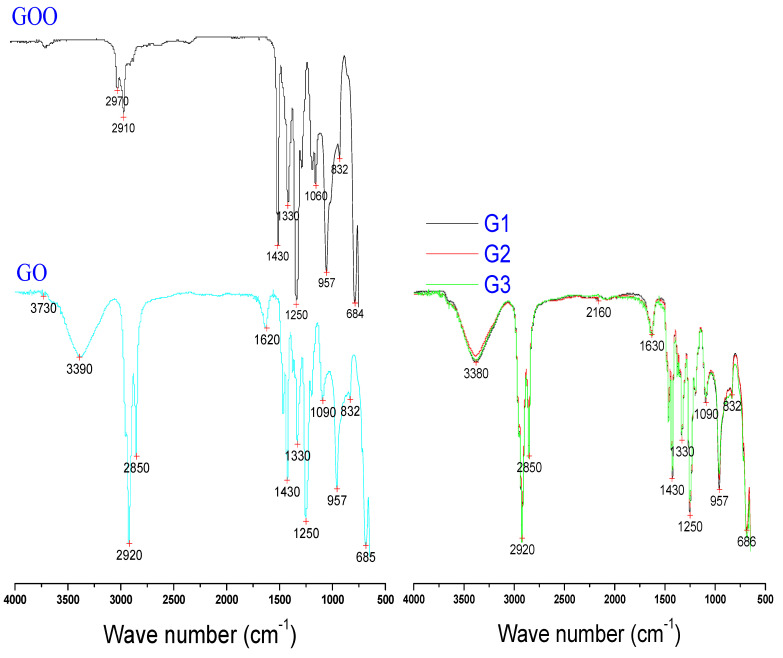
FTIR spectra of PIMs and their identical aligned constituents.

**Figure 4 membranes-12-00024-f004:**
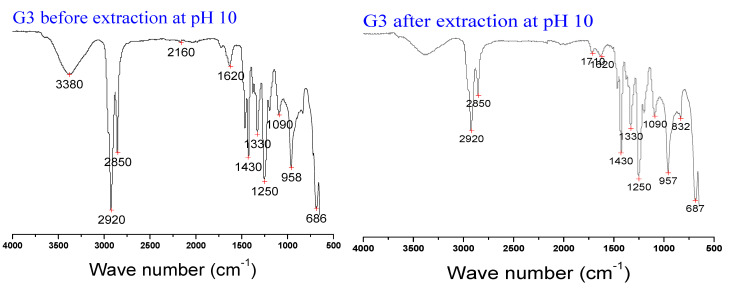
Compared FTIR spectra of G3 before and after Ibuprofen extraction.

**Figure 5 membranes-12-00024-f005:**
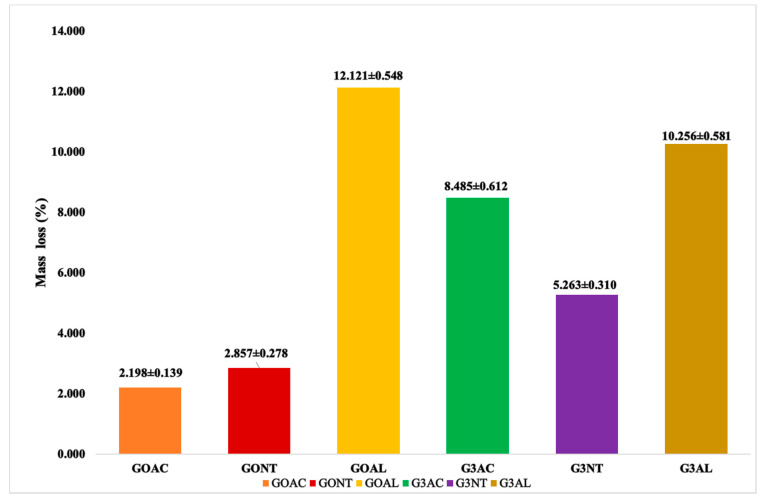
Influence of pH medium on the stability of graphene oxide doped PIM (G3) and control PIM (GO) (where AC = acidic medium, NT = neutral medium, AL = alkaline medium).

**Figure 6 membranes-12-00024-f006:**
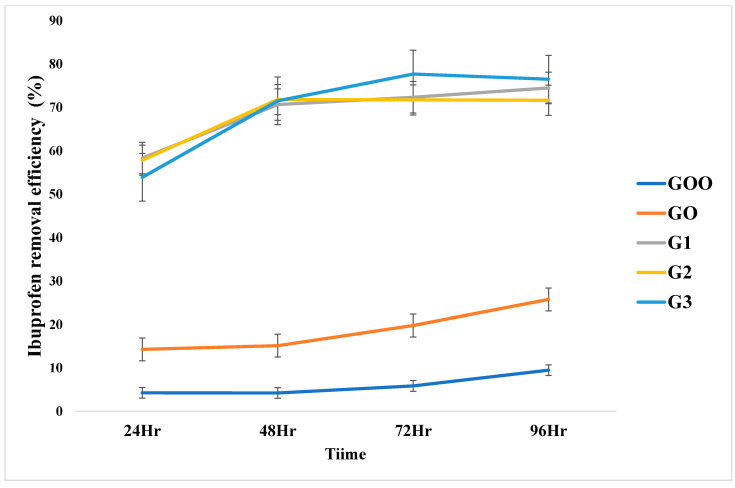
Effect of graphene concentration in the doped PIM on Ibuprofen extraction efficiency.

**Figure 7 membranes-12-00024-f007:**
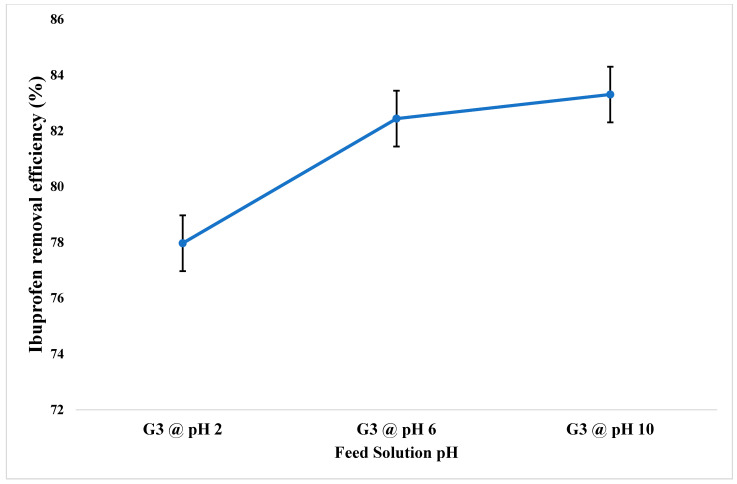
Effect of the pH of the feed solution on Ibuprofen extraction efficiency.

**Table 1 membranes-12-00024-t001:** Formulation Composition of synthesized PIMs.

PIM	PVC (g)	ALIQUAT (g)	THF (g)	GO (g)	% GO
GOO	6	-	24	-	-
GO	6	3	21	-	-
G1	6	3	21	0.045	0.15
G2	6	3	21	0.135	0.45
G3	6	3	21	0.225	0.75

**Table 2 membranes-12-00024-t002:** Physical properties of the pristine, control, and graphene oxide-doped PIMs.

PIM	Flexibility	Transparency
GOO	brittle	transparent
GO	brittle	transparent
G1	flexible	transparent
G2	flexible	transparent
G3	flexible	transparent

## Data Availability

Not applicable.

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
