# Peer review of "Graphene Oxide-Doped Polymer Inclusion Membrane for Remediation of Pharmaceutical Contaminant of Emerging Concerns: Ibuprofen"

_membranes, 2021, doi:10.3390/membranes12010024_

Round 1

Reviewer 1 Report

Comments

The subject matter is suitable for publication in Membranes.

I have read the article very carefully and it is, in general, an interesting paper because the authors report the results about the investigation over the efficiency and stability of graphene oxide doped polymer inclusion for the removal of low concentration Ibuprofen.

The fabrication and characterization of Graphene oxide (GO) doped PIMs has been developed. The membranes have been applied to extract aqueous Ibuprofen at varied pH conditions. The authors described that the doped PIMs membranes were synthesized using low concentration aliquat as carrier and 0, 0.15, 0.45, and 0.75 % GO as nanoparticles in a Polyvinyl chloride (PVC) base polymer without adding any plasticizer. The synthesized polymer inclusion membranes were characterized based on SEM, Ftir, physical and chemical stability.

The different sections of the manuscript are clear and well-structured. The authors compare, in some sections, their results with those of other experimental investigations and discuss them comparatively.

To improve the understanding of the manuscript, abbreviations section should be added.

Anyway, I include some specific comments:

  1. Materials and Methods

Have the authors reproduced these assays? Which is the reproducibility of these assays?

The authors should provide some data about the reproducibility of the experiments.

Some corrections about format

Sometimes Ibuprofen is written with capital letters and other times it is not. Please, check the format.

In page 2, line 56, authors say “among others found in GO. . These  

There are two points. Please check the paragraph and correct the mistake.

In page 4, line 149, “(SEM). scanning electron microscopy (SEM.) H.I.T.A.C.H.I. (TM3000.). The membrane  “

Please remove the points in the sentence and the word SEM is duplicated!

In page 6, 221 “was also indicated by suggests that the dispersed GO dispersion within  “

Please correct the duplicated expression.

In page 7, line 261, “ of the GO as indicated by the peak at 2160 cm− [48] (Tyagi et al., 2018)”

Please, add the correct units (cm-1).

Page 8, line 273, “A new distinct peak at 1710 cm 1

Idem, please check and correct the units!

In the same page, Table 2, remove the point extra in GO.

Page 9, line 310, “traction Ftir result ( Figure 5.). Nanoparticle loss du

Please check and correct, remove the point after Figure 5.

Author Response

Thanks so much for your time going through our manuscript and your valuable suggestions and observations for improvement. Kindly find attached our response to the issues raised for your perusal

Reviewer 2 Report

The paper is very well written.

  • In the Introduction, more info about wide range of contaminants presents in pharmaceutical WW (https://doi.org/10.1016/j.watres.2014.06.018) should be provided.
  • In my opinion, figure 1 is not necessary.
  • Sections 2.3.2.1. and 2.3.2.2. could be merged.
  • Resolution of Figure 2 is very low. Can you enhance it?
  • In figures 7 and 8, please, insert confidence intervals.
  • Conclusions should be implemented also with “numerical” results of your study.

Author Response

Kindly find attached our response to your helpful and engaging suggestions for improvement. 

Reviewer 3 Report

The manuscript details the use of a membrane doped with graphene oxide on the remediation of ibuprofen-contaminated water. It is an interesting read and it fits within the scope of the journal. Nonetheless, there are some corrections to be made prior to being publishable.

1) The manuscript needs some language checking. There are typos and grammatical inconsistencies all over it, and a professional editing service would be appropriate prior to resubmission.

2) Please check the reference style for the journal. The authors did not follow the format required by Membranes.

3) Line 20: FTIR, not Ftir

4) The figure depicting the ibuprofen structure is not needed in the text. Remove it.

5) The SEM micrographs are pretty bad. Are they really at a x20 amplification? If so, why did you use SEM and not another type of microscopy?

6) Figures 5 through 8 need thorough improvements. There are different styles, and they are very unappealing visually speaking. Please make them consistent and better looking.

Author Response

Thanks for sparing your time to give important hints and instructions to improve our manuscript. Herewith attached is the response to your comments.
